# CERTIFIED DEFENSES AGAINST ADVERSARIAL EXAMPLES

**Aditi Raghunathan, Jacob Steinhardt & Percy Liang**
Department of Computer Science
Stanford University
`{aditir,jsteinhardt,pliang}@cs.stanford.edu`

## ABSTRACT

While neural networks have achieved high accuracy on standard image classification benchmarks, their accuracy drops to nearly zero in the presence of small adversarial perturbations to test inputs. Defenses based on regularization and adversarial training have been proposed, but often followed by new, stronger attacks that defeat these defenses. Can we somehow end this arms race? In this work, we study this problem for neural networks with one hidden layer. We first propose a method based on a semidefinite relaxation that outputs a *certificate* that for a given network and test input, no attack can force the error to exceed a certain value. Second, as this certificate is differentiable, we jointly optimize it with the network parameters, providing an *adaptive regularizer* that encourages robustness against all attacks. On MNIST, our approach produces a network and a certificate that *no attack* that perturbs each pixel by at most $\epsilon = 0.1$ can cause more than $35\%$ test error.

## 1 INTRODUCTION

Despite the impressive (and sometimes even superhuman) accuracies of machine learning on diverse tasks such as object recognition (He et al., 2015), speech recognition (Xiong et al., 2016), and playing Go (Silver et al., 2016), classifiers still fail catastrophically in the presence of small imperceptible but *adversarial* perturbations (Szegedy et al., 2014; Goodfellow et al., 2015; Kurakin et al., 2016). In addition to being an intriguing phenonemon, the existence of such "adversarial examples" exposes a serious vulnerability in current ML systems (Evtimov et al., 2017; Sharif et al., 2016; Carlini et al., 2016). While formally defining an "imperceptible" perturbation is difficult, a commonly-used proxy is perturbations that are bounded in $\ell_\infty$-norm (Goodfellow et al., 2015; Madry et al., 2017; Tramèr et al., 2017); we focus on this attack model in this paper, as even for this proxy it is not known how to construct high-performing image classifiers that are robust to perturbations.

While a proposed defense (classifier) is often empirically shown to be successful against the set of attacks known at the time, new stronger attacks are subsequently discovered that render the defense useless. For example, defensive distillation (Papernot et al., 2016c) and adversarial training against the Fast Gradient Sign Method (Goodfellow et al., 2015) were two defenses that were later shown to be ineffective against stronger attacks (Carlini & Wagner, 2016; Tramèr et al., 2017). In order to break this *arms race* between attackers and defenders, we need to come up with defenses that are successful against *all attacks* within a certain class.

However, even computing the worst-case error for a given network against all adversarial perturbations in an $\ell_\infty$-ball is computationally intractable. One common approximation is to replace the worst-case loss with the loss from a given heuristic attack strategy, such as the Fast Gradient Sign Method (Goodfellow et al., 2015) or more powerful iterative methods (Carlini & Wagner, 2017a; Madry et al., 2017). *Adversarial training* minimizes the loss with respect to these heuristics. However, this essentially minimizes a *lower bound* on the worst-case loss, which is problematic since points where the bound is loose have disproportionately lower objective values, which could lure and mislead an optimizer. Indeed, while adversarial training often provides robustness against a specific attack, it often fails to generalize to new attacks, as described above. Another approach is to compute the worst-case perturbation *exactly* using discrete optimization (Katz et al., 2017a; Carlini

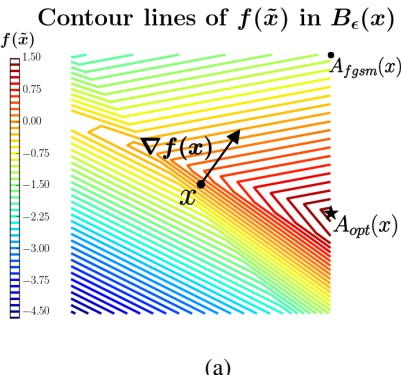
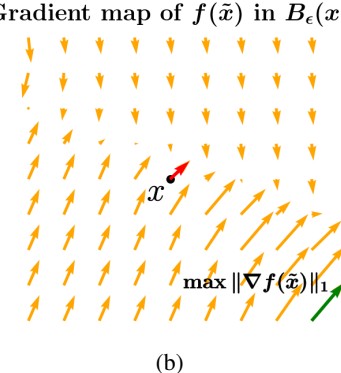

(a)  (b)

Figure 1: Illustration of the margin function $f(x)$ for a simple two-layer network. (a) Contours of $f(x)$ in an $\ell_\infty$ ball around $x$. Sharp curvature near $x$ renders a linear approximation highly inaccurate, and $f(A_{\text{fgsm}}(x))$ obtained by maximising this approximation is much smaller than $f(A_{\text{opt}}(x))$. (b) Vector field for $\nabla f(x)$ with length of arrows proportional to $\|\nabla f(x)\|_1$. In our approach, we bound $f(A_{\text{opt}}(x))$ by bounding the maximum of $\|\nabla f(\tilde{x})\|_1$ over the neighborhood (green arrow). In general, this could be very different from $\|\nabla f(x)\|_1$ at just the point $x$ (red arrow).

et al., 2017). Currently, these approaches can take up to several hours or longer to compute the loss for a single example even for small networks with a few hundred hidden units. Training a network would require performing this computation in the inner loop, which is infeasible.

In this paper, we introduce an approach that avoids both the inaccuracy of lower bounds and the intractability of exact computation, by computing an *upper bound* on the worst-case loss for neural networks with one hidden layer, based on a semidefinite relaxation that can be computed efficiently. This upper bound serves as a *certificate of robustness* against all attacks for a given network and input. Minimizing an upper bound is safer than minimizing a lower bound, because points where the bound is loose have disproportionately higher objective values, which the optimizer will tend to avoid. Furthermore, our certificate of robustness, by virtue of being differentiable, is *trainable*—it can be optimized at training time jointly with the network, acting as a regularizer that encourages robustness against all $\ell_\infty$ attacks.

In summary, we are the first (along with the concurrent work of Kolter & Wong (2017)) to demonstrate a certifiable, trainable, and scalable method for defending against adversarial examples on two-layer networks. We train a network on MNIST whose test error on clean data is $4.2\%$, and which comes with a certificate that no attack can misclassify more than $35\%$ of the test examples using $\ell_\infty$ perturbations of size $\epsilon = 0.1$.

**Notation.** For a vector $z \in \mathbb{R}^n$, we use $z_i$ to denote the $i^{\text{th}}$ coordinate of $z$. For a matrix $Z \in \mathbb{R}^{m \times n}$, $Z_i$ denotes the $i^{\text{th}}$ row. For any activation function $\sigma : \mathbb{R} \to \mathbb{R}$ (e.g., sigmoid, ReLU) and a vector $z \in \mathbb{R}^n$, $\sigma(z)$ is a vector in $\mathbb{R}^n$ with $\sigma(z)_i = \sigma(z_i)$ (non-linearity is applied element-wise). We use $B_\epsilon(z)$ to denote the $\ell_\infty$ ball of radius $\epsilon$ around $z \in \mathbb{R}^d$: $B_\epsilon(z) = \{\tilde{z} \mid |\tilde{z}_i - z_i| \leq \epsilon \text{ for } i = 1, 2, \ldots d\}$. Finally, we denote the vector of all zeros by $\mathbf{0}$ and the vector of all ones by $\mathbf{1}$.

## 2 SETUP

**Score-based classifiers.** Our goal is to learn a mapping $C : \mathcal{X} \to \mathcal{Y}$, where $\mathcal{X} = \mathbb{R}^d$ is the input space (e.g., images) and $\mathcal{Y} = \{1, \ldots, k\}$ is the set of $k$ class labels (e.g., object categories). Assume $C$ is driven by a scoring function $f^i : \mathcal{X} \to \mathbb{R}$ for all classes $i \in \mathcal{Y}$, where the classifier chooses the class with the highest score: $C(x) = \arg\max_{i \in \mathcal{Y}} f^i(x)$. Also, define the *pairwise margin* $f^{ij}(x) \stackrel{\text{def}}{=} f^i(x) - f^j(x)$ for every pair of classes $(i, j)$. Note that the classifier outputs $C(x) = i$ iff $f^{ij}(x) > 0$ for all alternative classes $j \neq i$. Normally, a classifier is evaluated on the 0-1 loss $\ell(x, y) = \mathbb{I}[C(x) \neq y]$.

This paper focuses on linear classifiers and neural networks with one hidden layer. For linear classifiers, $f^i(x) \stackrel{\text{def}}{=} W_i^\top x$, where $W_i$ is the $i^{\text{th}}$ row of the parameter matrix $W \in \mathbb{R}^{k \times d}$.

For neural networks with one hidden layer consisting of $m$ hidden units, the scoring function is $f^i(x) = V_i^\top \sigma(Wx)$, where $W \in \mathbb{R}^{m \times d}$ and $V \in \mathbb{R}^{k \times m}$ are the parameters of the first and second layer, respectively, and $\sigma$ is a non-linear activation function applied elementwise (e.g., for ReLUs, $\sigma(z) = \max(z, 0)$). We will assume below that the gradients of $\sigma$ are bounded: $\sigma'(z) \in [0, 1]$ for all $z \in \mathbb{R}$; this is true for ReLUs, as well as for sigmoids (with the stronger bound $\sigma'(z) \in [0, \frac{1}{4}]$).

**Attack model.** We are interested in classification in the presence of an attacker $A : \mathcal{X} \to \mathcal{X}$ that takes a (test) input $x$ and returns a perturbation $\tilde{x}$. We consider attackers $A$ that can perturb each feature $x_i$ by at most $\epsilon \geq 0$; formally, $A(x)$ is required to lie in the $\ell_\infty$ ball $B_\epsilon(x) \overset{\text{def}}{=} \{\tilde{x} \mid \|\tilde{x} - x\|_\infty \leq \epsilon\}$, which is the standard constraint first proposed in Szegedy et al. (2014). Define the *adversarial loss* with respect to $A$ as $\ell_A(x, y) = \mathbb{I}[C(A(x)) \neq y]$.

We assume the white-box setting, where the attacker $A$ has full knowledge of $C$. The optimal (untargeted) attack chooses the input that maximizes the pairwise margin of an incorrect class $i$ over the correct class $y$: $A_{\text{opt}}(x) = \arg\max_{\tilde{x} \in B_\epsilon(x)} \max_i f^{iy}(\tilde{x})$. For a neural network, computing $A_{\text{opt}}$ is a non-convex optimization problem; heuristics are typically employed, such as the Fast Gradient Sign Method (FGSM) (Goodfellow et al., 2015), which perturbs $x$ based on the gradient, or the Carlini-Wagner attack, which performs iterative optimization (Carlini & Wagner, 2017b).

## 3 Certificate on the adversarial loss

For ease of exposition, we first consider binary classification with classes $\mathcal{Y} = \{1, 2\}$; the multiclass extension is discussed at the end of Section 3.3. Without loss of generality, assume the correct label for $x$ is $y = 2$. Simplifying notation, let $f(x) = f^1(x) - f^2(x)$ be the margin of the incorrect class over the correct class. Then $A_{\text{opt}}(x) = \arg\max_{\tilde{x} \in B_\epsilon(x)} f(\tilde{x})$ is the optimal attack, which is successful if $f(A_{\text{opt}}(x)) > 0$. Since $f(A_{\text{opt}}(x))$ is intractable to compute, we will try to upper bound it via a tractable relaxation.

In the rest of this section, we first review a classic result in the simple case of linear networks where a tight upper bound is based on the $\ell_1$-norm of the weights (Section 3.1). We then extend this to general classifiers, in which $f(A_{\text{opt}}(x))$ can be upper bounded using the maximum $\ell_1$-norm of the gradient at any point $\tilde{x} \in B_\epsilon(x)$ (Section 3.2). For two-layer networks, this quantity is upper bounded by the optimal value $f_{\text{QP}}(x)$ of a non-convex quadratic program (QP) (Section 3.3), which in turn is upper bounded by the optimal value $f_{\text{SDP}}(x)$ of a semidefinite program (SDP). The SDP is convex and can be computed exactly (which is important for obtainining actual certificates). To summarize, we have the following chain of inequalities:

$$f(A(x)) \leq f(A_{\text{opt}}(x)) \overset{(3.2)}{\leq} f(x) + \epsilon \max_{\tilde{x} \in B_\epsilon(x)} \|\nabla f(\tilde{x})\|_1 \overset{(3.3)}{\leq} f_{\text{QP}}(x) \overset{(3.3)}{\leq} f_{\text{SDP}}(x), \qquad (1)$$

which implies that the adversarial loss $\ell_A(x) = \mathbb{I}[f(A(x)) > 0]$ with respect to any attacker $A$ is upper bounded by $\mathbb{I}[f_{\text{SDP}}(x) > 0]$. Note that for certain non-linearities such as ReLUs, $\nabla f(\tilde{x})$ does not exist everywhere, but our analysis below holds as long as $f$ is differentiable almost-everywhere.

### 3.1 Linear classifiers

For (binary) linear classifiers, we have $f(x) = (W_1 - W_2)^\top x$, where $W_1, W_2 \in \mathbb{R}^d$ are the weight vectors for the two classes. For any input $\tilde{x} \in B_\epsilon(x)$, Hölder's inequality with $\|x - \tilde{x}\|_\infty \leq \epsilon$ gives:

$$f(\tilde{x}) = f(x) + (W_1 - W_2)^\top (\tilde{x} - x) \leq f(x) + \epsilon \|W_1 - W_2\|_1. \qquad (2)$$

Note that this bound is tight, obtained by taking $A_{\text{opt}}(x)_i = x_i + \epsilon \operatorname{sign}(W_{1i} - W_{2i})$.

### 3.2 General classifiers

For more general classifiers, we cannot compute $f(A_{\text{opt}}(x))$ exactly, but motivated by the above, we can use the gradient to obtain a *linear approximation* $g$:

$$f(\tilde{x}) \approx g(\tilde{x}) \overset{\text{def}}{=} f(x) + \nabla f(x)^\top (\tilde{x} - x) \leq f(x) + \epsilon \|\nabla f(x)\|_1. \qquad (3)$$

Using this linear approximation to generate $A(x)$ corresponds exactly to the Fast Gradient Sign Method (FGSM) (Goodfellow et al., 2015). However, $f$ is only close to $g$ when $\tilde{x}$ is very close to $x$, and people have observed the *gradient masking* phenomenon (Tramèr et al., 2017; Papernot et al., 2016b) in several proposed defenses that train against approximations like $g$, such as saturating networks (Nayebi & Ganguli, 2017), distillation (Papernot et al., 2016c), and adversarial training (Goodfellow et al., 2015). Specifically, defenses that try to minimize $\|\nabla f(x)\|_1$ locally at the training points result in loss surfaces that exhibit sharp curvature near those points, essentially rendering the linear approximation $g(\tilde{x})$ meaningless. Some attacks (Carlini & Wagner, 2016; Tramèr et al., 2017) evade these defenses and witness a large $f(A_{\text{opt}}(x))$. Figure 1a provides a simple illustration.

We propose an alternative approach: use integration to obtain an *exact* expression for $f(\tilde{x})$ in terms of the gradients along the line between $x$ and $\tilde{x}$:

$$
\begin{aligned}
f(\tilde{x}) &= f(x) + \int_0^1 \nabla f\big(t\tilde{x} + (1-t)x\big)^\top (\tilde{x} - x)\, dt \\
&\leq f(x) + \max_{\tilde{x} \in B_\epsilon(x)} \epsilon \|\nabla f(\tilde{x})\|_1,
\end{aligned} \tag{4}
$$

where the inequality follows from the fact that $t\tilde{x} + (1-t)x \in B_\epsilon(x)$ for all $t \in [0,1]$. The key difference between (4) and (3) is that we consider the gradients over the entire ball $B_\epsilon(x)$ rather than only at $x$ (Figure 1b). However, computing the RHS of (4) is intractable in general. For two-layer neural networks, this optimization has additional structure which we will exploit in the next section.

### 3.3 TWO-LAYER NEURAL NETWORKS

We now unpack the upper bound (4) for two-layer neural networks. Recall from Section 2 that $f(x) = f^1(x) - f^2(x) = v^\top \sigma(Wx)$, where $v \overset{\text{def}}{=} V_1 - V_2 \in \mathbb{R}^m$ is the difference in second-layer weights for the two classes. Let us try to bound the norm of the gradient $\|\nabla f(\tilde{x})\|_1$ for $\tilde{x} \in B_\epsilon(x)$. If we apply the chain rule, we see that the only dependence on $\tilde{x}$ is $\sigma'(W\tilde{x})$, the activation derivatives. We now leverage our assumption that $\sigma'(z) \in [0,1]^m$ for all vectors $z \in \mathbb{R}^m$, so that we can optimize over possible activation derivatives $s \in [0,1]^m$ directly *independent of* $x$ (note that there is potential looseness because not all such $s$ need be obtainable via some $\tilde{x} \in B_\epsilon(x)$). Therefore:

$$
\begin{aligned}
\|\nabla f(\tilde{x})\|_1 &\overset{(i)}{=} \|W^\top \operatorname{diag}(v)\sigma'(W\tilde{x})\|_1 \\
&\overset{(ii)}{\leq} \max_{s \in [0,1]^m} \|W^\top \operatorname{diag}(v)s\|_1 \\
&\overset{(iii)}{=} \max_{s \in [0,1]^m, t \in [-1,1]^d} t^\top W^\top \operatorname{diag}(v)s,
\end{aligned} \tag{5}
$$

where (i) follows from the chain rule, (ii) uses the fact that $\sigma$ has bounded derivatives $\sigma'(z) \in [0,1]$, and (iii) follows from the identity $\|z\|_1 = \max_{t \in [-1,1]^d} t^\top z$. (Note that for sigmoid networks, where $\sigma'(z) \in [0, \frac{1}{4}]$, we can strengthen the above bound by a corresponding factor of $\frac{1}{4}$.) Substituting the bound (5) into (4), we obtain an upper bound on the adversarial loss that we call $f_{\text{QP}}$:

$$
\begin{aligned}
f(A_{\text{opt}}(x)) &\leq f(x) + \epsilon \max_{\tilde{x} \in B_\epsilon(x)} \|\nabla f(\tilde{x})\|_1 \\
&\leq f(x) + \epsilon \max_{s \in [0,1]^m, t \in [-1,1]^d} t^\top W^\top \operatorname{diag}(v)s \overset{\text{def}}{=} f_{\text{QP}}(x).
\end{aligned} \tag{6}
$$

Unfortunately, (6) still involves a non-convex optimization problem (since $W^\top \operatorname{diag}(v)$ is not necessarily negative semidefinite). In fact, it is similar to the NP-hard MAXCUT problem, which requires maximizing $x^\top L x$ over $x \in [-1,1]^d$ for a graph with Laplacian matrix $L$.

While MAXCUT is NP-hard, it can be efficiently approximated, as shown by the celebrated semidefinite programming relaxation for MAXCUT in Goemans & Williamson (1995). We follow a similar approach here to obtain an upper bound on $f_{\text{QP}}(x)$.

First, to make our variables lie in $[-1,1]^m$ instead of $[0,1]^m$, we reparametrize $s$ to produce:

$$
\max_{s \in [-1,1]^m, t \in [-1,1]^d} \frac{1}{2} t^\top W^\top \operatorname{diag}(v)(\mathbf{1} + s). \tag{7}
$$

Next pack the variables into a vector $y \in \mathbb{R}^{m+d+1}$ and the parameters into a matrix $M$:

$$y \stackrel{\text{def}}{=} \begin{bmatrix} 1 \\ t \\ s \end{bmatrix} \quad M(v, W) \stackrel{\text{def}}{=} \begin{bmatrix} 0 & 0 & \mathbf{1}^\top W^\top \operatorname{diag}(v) \\ 0 & 0 & W^\top \operatorname{diag}(v) \\ \operatorname{diag}(v)^\top W \mathbf{1} & \operatorname{diag}(v)^\top W & 0 \end{bmatrix}. \quad (8)$$

In terms of these new objects, our objective takes the form:

$$\max_{y \in [-1,1]^{(m+d+1)}} \frac{1}{4} y^\top M(v, W) y = \max_{y \in [-1,1]^{(m+d+1)}} \frac{1}{4} \langle M(v, W), yy^\top \rangle. \quad (9)$$

Note that every valid vector $y \in [-1, +1]^{m+d+1}$ satisfies the constraints $yy^\top \succeq 0$ and $(yy^\top)_{jj} = 1$. Defining $P = yy^\top$, we obtain the following *convex* semidefinite relaxation of our problem:

$$f_{\text{QP}}(x) \le f_{\text{SDP}}(x) \stackrel{\text{def}}{=} f(x) + \frac{\epsilon}{4} \max_{P \succeq 0, \operatorname{diag}(P) \le 1} \langle M(v, W), P \rangle. \quad (10)$$

Note that the optimization of the semidefinite program depends only on the weights $v$ and $W$ and does not depend on the inputs $x$, so it only needs to be computed once for a model $(v, W)$.

Semidefinite programs can be solved with off-the-shelf optimizers, although these optimizers are somewhat slow on large instances. In Section 4 we propose a fast stochastic method for training, which only requires computing the top eigenvalue of a matrix.

**Generalization to multiple classes.** The preceding arguments all generalize to the pairwise margins $f^{ij}$, to give:

$$f^{ij}(A(x)) \le f^{ij}_{\text{SDP}}(x) \stackrel{\text{def}}{=} f^{ij}(x) + \frac{\epsilon}{4} \max_{P \succeq 0, \operatorname{diag}(P) \le 1} \langle M^{ij}(V, W), P \rangle, \text{ where} \quad (11)$$

$M^{ij}(V, W)$ is defined as in (9) with $v = V_i - V_j$. The adversarial loss of any attacker, $\ell_A(x, y) = \mathbb{I}[\max_{i \ne y} f^{iy}(A(x)) > 0]$, can be bounded using the fact that $f^{iy}_{\text{SDP}}(x) \ge f^{iy}(A(x))$. In particular,

$$\ell_A(x, y) = 0 \text{ if } \max_{i \ne y} f^{iy}_{\text{SDP}}(x) < 0. \quad (12)$$

## 4 TRAINING THE CERTIFICATE

In the previous section, we proposed an upper bound (12) on the loss $\ell_A(x, y)$ of any attack A, based on the bound (11). Normal training with some classification loss $\ell_{\text{cls}}(V, W; x_n, y_n)$ like hinge loss or cross-entropy will encourage the *pairwise margin* $f^{ij}(x)$ to be large in magnitude, but won't necessarily cause the second term in (11) involving $M^{ij}$ to be small. A natural strategy is thus to use the following regularized objective given training examples $(x_n, y_n)$, which pushes down on both terms:

$$(W^\star, V^\star) = \underset{W,V}{\arg\min} \sum_n \ell_{\text{cls}}(V, W; x_n, y_n) + \sum_{i \ne j} \lambda^{ij} \max_{P \succeq 0, \operatorname{diag}(P) \le 1} \langle M^{ij}(V, W), P \rangle, \quad (13)$$

where $\lambda^{ij} > 0$ are the regularization hyperparameters. However, computing the gradients of the above objective involves finding the optimal solution of a semidefinite program, which is slow.

**Duality to the rescue.** Our computational burden is lifted by the beautiful theory of duality, which provides the following equivalence between the *primal* maximization problem over $P$, and a *dual* minimization problem over new variables $c$ (see Section A for details):

$$\max_{P \succeq 0, \operatorname{diag}(P) \le 1} \langle M^{ij}(V, W), P \rangle = \min_{c^{ij} \in \mathbf{R}^D} D \cdot \lambda^+_{\max}\left(M^{ij}(V, W) - \operatorname{diag}(c^{ij})\right) + \mathbf{1}^\top \max(c, 0), \quad (14)$$

where $D = (d + m + 1)$ and $\lambda^+_{\max}(B)$ is the maximum eigenvalue of $B$, or 0 if all eigenvalues are negative. This dual formulation allows us to introduce additional dual variables $c^{ij} \in \mathbb{R}^D$ that are optimized *at the same time as the parameters $V$ and $W$*, resulting in an objective that can be trained efficiently using stochastic gradient methods.

**The final objective.** Using (14), we end up optimizing the following training objective:

$$(W^\star, V^\star, c^\star) = \underset{W,V,c}{\arg\min} \sum_n \ell_{\text{cls}}(V, W; x_n, y_n) + \sum_{i \ne j} \lambda^{ij} \cdot \left[D \cdot \lambda^+_{\max}(M^{ij}(V, W) - \operatorname{diag}(c^{ij})) + \mathbf{1}^\top \max(c^{ij}, 0)\right].$$

$$(15)$$

The objective in (15) can be optimized efficiently. The most expensive operation is $\lambda_{\max}^+$, which requires computing the maximum eigenvector of the matrix $M^{ij} - \mathrm{diag}(c^{ij})$ in order to take gradients. This can be done efficiently using standard implementations of iterative methods like Lanczos. Further implementation details (including tuning of $\lambda^{ij}$) are presented in Section 6.3.

**Dual certificate of robustness.** The dual formulation is also useful because *any* value of the dual is an upper bound on the optimal value of the primal. Specifically, if $(W[t], V[t], c[t])$ are the parameters at iteration $t$ of training, then

$$f^{ij}(A(x)) \leq f(x) + \frac{\epsilon}{4} \left[ D \cdot \lambda_{\max}^+ \big( M^{ij}(V[t], W[t]) - \mathrm{diag}(c[t]^{ij}) \big) + \mathbf{1}^\top \max(c[t]^{ij}, 0) \right], \quad (16)$$

for any attack $A$. As we train the network, we obtain a quick upper bound on the worst-case adversarial loss *directly from the regularization loss*, without having to optimize an SDP each time.

## 5 OTHER UPPER BOUNDS

In Section 3, we described a function $f_{\mathrm{SDP}}^{ij}$ that yields an efficient upper bound on the adversarial loss, which we obtained using convex relaxations. One could consider other simple ways to upper bound the loss; we describe here two common ones based on the spectral and Frobenius norms.

**Spectral bound:** Note that $v^\top(\sigma(W\tilde{x}) - \sigma(Wx)) \leq \|v\|_2 \|\sigma(W\tilde{x}) - \sigma(Wx)\|_2$ by Cauchy-Schwarz. Moreover, since $\sigma$ is contractive, $\|\sigma(W\tilde{x}) - \sigma(Wx)\|_2 \leq \|W(\tilde{x} - x)\|_2 \leq \|W\|_2\|\tilde{x} - x\|_2 \leq \epsilon\sqrt{d}\|W\|_2$, where $\|W\|_2$ is the *spectral norm* (maximum singular value) of $W$. This yields the following upper bound that we denote by $f_{\mathrm{spectral}}$:

$$f^{ij}(A(x)) \leq f_{\mathrm{spectral}}^{ij}(x) \overset{\mathrm{def}}{=} f^{ij}(x) + \epsilon\sqrt{d}\|W\|_2\|V_i - V_j\|_2. \quad (17)$$

This measure of vulnerability to adversarial examples based on the spectral norms of the weights of each layer is considered in Szegedy et al. (2014) and Cisse et al. (2017).

**Frobenius bound:** For ease in training, often the Frobenius norm is regularized (weight decay) instead of the spectral norm. Since $\|W\|_{\mathrm{F}} \geq \|W\|_2$, we get a corresponding upper bound $f_{\mathrm{frobenius}}$:

$$f^{ij}(A(x)) \leq f_{\mathrm{frobenius}}^{ij}(x) = f^{ij}(x) + \epsilon\sqrt{d}\|W\|_{\mathrm{F}}\|V_i - V_j\|_2. \quad (18)$$

In Section 6, we empirically compare our proposed bound using $f_{\mathrm{SDP}}^{ij}$ to these two upper bounds.

## 6 EXPERIMENTS

We evaluated our method on the MNIST dataset of handwritten digits, where the task is to classify images into one of ten classes. Our results can be summarized as follows: First, in Section 6.1, we show that our certificates of robustness are tighter than those based on simpler methods such as Frobenius and spectral bounds (Section 5), but our bounds are still too high to be meaningful for general networks. Then in Section 6.2, we show that by training on the certificates, we obtain networks with much better bounds and hence meaningful robustness. This reflects an important point: while accurately analyzing the robustness of an arbitrary network is hard, training the certificate jointly leads to a network that is robust and certifiably so. In Section 6.3, we present implementation details, design choices, and empirical observations that we made while implementing our method.

**Networks.** In this work, we focus on two layer networks. In all our experiments, we used neural networks with $m = 500$ hidden units, and TensorFlow's implementation of Adam (Kingma & Ba, 2014) as the optimizer; we considered networks with more hidden units, but these did not substantially improve accuracy. We experimented with both the multiclass hinge loss and cross-entropy. All hyperparameters (including the choice of loss function) were tuned based on the error of the Projected Gradient Descent (PGD) attack (Madry et al., 2017) at $\epsilon = 0.1$; we report the hyperparameter settings below. We considered the following training objectives providing 5 different networks:

1. **Normal training (NT-NN).** Cross-entropy loss and no explicit regularization.
2. **Frobenius norm regularization (Fro-NN).** Hinge loss and a regularizer $\lambda(\|W\|_{\mathrm{F}} + \|v\|_2)$ with $\lambda = 0.08$.

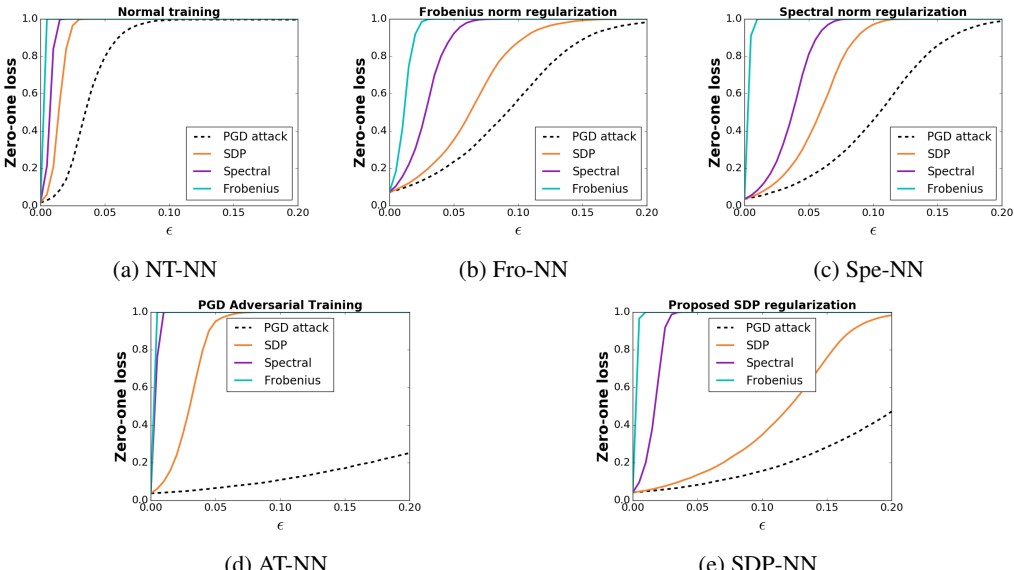

Figure 2: Upper bounds on adversarial error for different networks on MNIST.

3. **Spectral norm regularization (Spe-NN).** Hinge loss and a regularizer $\lambda(\|W\|_2 + \|v\|_2)$ with $\lambda = 0.09$.

4. **Adversarial training (AT-NN).** Cross-entropy with the adversarial loss against PGD as a regularizer, with the regularization parameter set to $0.5$. We found that this regularized loss works better than optimizing only the adversarial loss, which is the defense proposed in Madry et al. (2017). We set the step size of the PGD adversary to $0.1$, number of iterations to $40$, and perturbation size to $0.3$.

5. **Proposed training objective (SDP-NN).** Dual SDP objective described in Equation 15 of Section 4. Implementation details and hyperparameter values are detailed in Section 6.3.

**Evaluating upper bounds.** Below we will consider various upper bounds on the adversarial loss $\ell_{A_{\mathrm{opt}}}$ (based on our method, as well as the Frobenius and spectral bounds described in Section 5). Ideally we would compare these to the ground-truth adversarial loss $\ell_{A_{\mathrm{opt}}}$, but computing this exactly is difficult. Therefore, we compare upper bounds on the adversarial loss with a *lower bound* on $\ell_{A_{\mathrm{opt}}}$ instead. The loss of any attack provides a valid lower bound and we consider the strong Projected Gradient Descent (PGD) attack run against the cross-entropy loss, starting from a random point in $B_\epsilon(x)$, with 5 random restarts. We observed that PGD against hinge loss did not work well, so we used cross-entropy even for attacking networks trained with the hinge loss.

## 6.1 QUALITY OF THE UPPER BOUND

For each of the five networks described above, we computed upper bounds on the 0-1 loss based on our certificate (which we refer to as the "SDP bound" in this section), as well as the Frobenius and spectral bounds described in Section 5. While Section 4 provides a procedure for efficiently obtaining an SDP bound as a result of training, for networks not trained with our method we need to solve an SDP at the end of training to obtain certificates. Fortunately, this only needs to be done *once* for every pair of classes. In our experiments, we use the modeling toolbox YALMIP (Löfberg, 2004) with Sedumi (Sturm, 1999) as a backend to solve the SDPs, using the dual form (14); this took roughly 10 minutes per SDP (around 8 hours in total for a given model).

In Figure 2, we display average values of the different upper bounds over the $10,000$ test examples, as well as the corresponding lower bound from PGD. We find that our bound is tighter than the Frobenius and spectral bounds for all the networks considered, but its tightness relative to the PGD lower bound varies across the networks. For instance, our bound is relatively tight on Fro-NN, but unfortunately Fro-NN is not very robust against adversarial examples (the PGD attack exhibits large

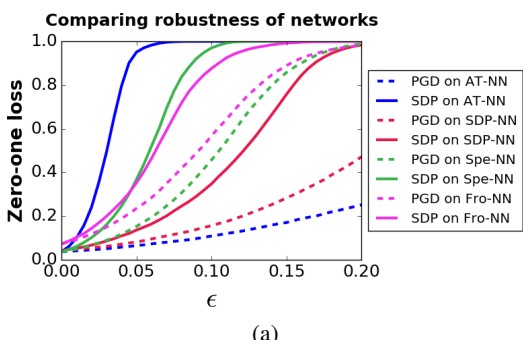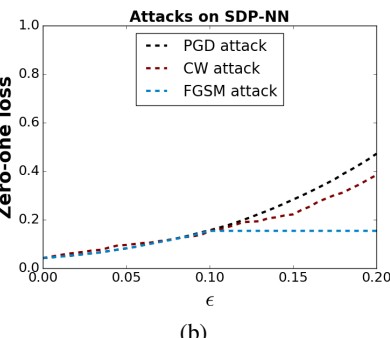

Figure 3: (a) Upper bound (SDP) and lower bound (PGD) on the adversarial error for different networks. (b) Error of SDP-NN against 3 different attacks.

error). In contrast, the adversarially trained network AT-NN does appear to be robust to attacks, but our certificate, despite being much tighter than the Frobenius and spectral bounds, is far away from the PGD lower bound. The only network that is both robust and has relatively tight upper bounds is SDP-NN, which was explicitly trained to be both robust and certifiable as described in Section 4; we examine this network and the effects of training in more detail in the next subsection.

## 6.2 EVALUATING PROPOSED TRAINING OBJECTIVE.

In the previous section, we saw that the SDP bound, while being tighter than simpler upper bounds, could still be quite loose on arbitrary networks. However, optimizing against the SDP certificate seemed to make the certificate tighter. In this section, we explore the effect of different optimization objectives in more detail. First, we plot on a single axis the best upper bound (i.e., the SDP bound) and the lower bound (from PGD) on the adversarial loss obtained with each of the five training objectives discussed above. This is given in Figure 3a.

Neither spectral nor Frobenius norm regularization seems to be helpful for encouraging adversarial robustness—the actual performance of those networks against the PGD attack is worse than the *upper bound* for SDP-NN against all attacks. This shows that the SDP certificate actually provides a useful training objective for encouraging robustness compared to other regularizers.

Separately, we can ask whether SDP-NN is robust to actual attacks. We explore the robustness of our network in Figure 3b, where we plot the performance of SDP-NN against 3 attacks—the PGD attack from before, the Carlini-Wagner attack (Carlini & Wagner, 2017b) (another strong attack), and the weaker Fast Gradient Sign Method (FGSM) baseline. We see substantial robustness against all 3 attacks, even though our method was not explicitly trained with any of them in mind.

Next, we compare to other bounds reported in the literature. A rough ceiling is given by the network of Madry et al. (2017), which is a relatively large four-layer convolutional network adversarially trained against PGD. While this network has no accompanying certificate of robustness, it was evaluated against a number of attack strategies and had worst-case error $11\%$ at $\epsilon = 0.3$. Another set of numbers comes from Carlini et al. (2017), who use formal verification methods to compute $A_{opt}$ exactly on 10 input examples for a small (72-node) variant of the Madry et al. network. The authors reported to us that this network misclassifies 6 out of 10 examples at $\epsilon = 0.05$ (we note that 4 out of 10 of these were misclassified to start with, but 3 of the 4 can also be flipped to a different wrong class with some $\epsilon < 0.07$).

At the value $\epsilon = 0.1$ for which it was tuned, SDP-NN has error $16\%$ against the PGD attack, and an upper bound of $35\%$ error against any attack. This is substantially better than the small 72-node network, but also much worse than the full Madry et al. network. How much of the latter looseness comes from conservatism in our method, versus the fact that our network has only two layers? We can get some idea by considering the AT-NN network, which was trained similarly to Madry et al., but uses the same architecture as SDP-NN. From Figure 3a, we see that the error of SDP-NN against PGD ($16\%$) is not much worse than that of AT-NN ($11\%$), even though AT-NN was explicitly trained against the PGD attack. This suggests that most of the gap comes from the smaller network depth,

| Network | PGD error | SDP bound | LP bound |
|---------|-----------|-----------|----------|
| SDP-NN | 15% | 35% | 99% |
| LP-NN | 22% | 93% | 26% |

Table 1: Comparison with the bound (LP bound) and training approach (LP-NN) of Kolter & Wong (2017). Numbers are reported for $\epsilon = 0.1$. LP-NN has a certificate (provided by the LP bound) that no attack can misclassify more than 26% of the examples.

rather than from conservatism in the SDP bound. We are currently in the process of extending our approach to deeper networks, and optimistic about obtaining improved bounds with such networks.

Finally, we compare with the approach proposed in Kolter & Wong (2017) whose work appeared shortly after an initial version of our paper. They provide an upper bound on the adversarial loss using linear programs (LP) followed by a method to efficiently train networks to minimize this upper bound. In order to compare with SDP-NN, the authors provided us with a network with the same architecture as SDP-NN, but trained using their LP based objective. We call this network LP-NN. Table 1 shows that LP-NN and SDP-NN are comparable in terms of their robustness against PGD, and the robustness guarantees that they come with.

Interestingly, the SDP and LP approaches provide vacuous bounds for networks *not* trained to minimize the respective upper bounds (though these networks are indeed robust). This suggests that these two approaches are comparable, but complementary. Finally, we note that in contrast to this work, the approach of Kolter & Wong (2017) extends to deeper networks, which allows them to train a four-layer CNN with a provable upper bound on adversarial error of 8.4% error.

### 6.3 IMPLEMENTATION DETAILS

We implemented our training objective in TensorFlow, and implemented $\lambda_{\max}^+$ as a custom operator using SciPy's implementation of the Lanczos algorithm for fast top eigenvector computation; occasionally Lanczos fails to converge due to a small eigen-gap, in which case we back off to a full SVD. We used hinge loss as the classification loss, and decayed the learning rate in steps from $10^{-3}$ to $10^{-5}$, decreasing by a factor of 10 every 30 epochs. Each gradient step involves computing top eigenvectors for 45 different matrices, one for each pair of classes $(i, j)$. In order to speed up computation, for each update, we randomly pick $i_t$ and only compute gradients for pairs $(i_t, j), j \neq i_t$, requiring only 9 top eigenvector computations in each step.

For the regularization parameters $\lambda^{ij}$, the simplest idea is to set them all equal to the same value; this leads to the **unweighted** regularization scheme where $\lambda^{ij} = \lambda$ for all pairs $(i, j)$. We tuned $\lambda$ to 0.05, which led to reasonably good bounds. However, we observed that certain pairs of classes tended to have larger margins $f^{ij}(x)$ than other classes, which meant that certain label pairs appeared in the maximum of (12) much more often. That led us to consider a **weighted** regularization scheme with $\lambda^{ij} = w^{ij}\lambda$, where $w^{ij}$ is the fraction of training points for which the the label $i$ (or $j$) appears as the maximizing term in (12). We updated the values of these weights every 20 epochs. Figure 4a compares the PGD lower bound and SDP upper bound for the unweighted and weighted networks. The weighted network is better than the unweighted network for both the lower and upper bounds.

Finally, we saw in Equation 16 of Section 4 that the dual variables $c^{ij}$ provide a quick-to-compute certificate of robustness. Figure 4b shows that the certificates provided by these dual variables are very close to what we would obtain by fully optimizing the semidefinite programs. These dual certificates made it easy to track robustness across epochs of training and to tune hyperparameters.

## 7 DISCUSSION

In this work, we proposed a method for producing *certificates of robustness* for neural networks, and for training against these certificates to obtain networks that are provably robust against adversaries.

**Related work.** In parallel and independent work, Kolter & Wong (2017) also provide provably robust networks against $\ell_\infty$ perturbations by using convex relaxations. While our approach uses a single semidefinite program to compute an upper bound on the adversarial loss, Kolter & Wong

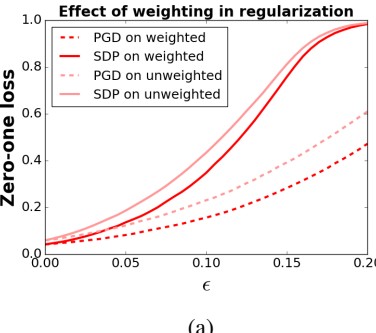 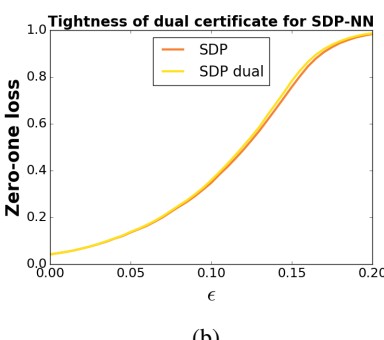

(a)                                                    (b)

Figure 4: (a) Weighted and unweighted regularization schemes. The network produced by weighting has a better certificate as well as lower error against the PGD attack. (b) The dual certificate of robustness (SDP dual), obtained automatically during training, is almost as good as the certificate produced by exactly solving the SDP.

(2017) use separate linear programs for every data point, and apply their method to networks of depth up to four. In theory, neither bound is strictly tighter than the other, and our experiments (Table 1) suggest that the two bounds are complementary. Combining the approaches seems to be a promising future direction.

Katz et al. (2017a) and the follow-up Carlini et al. (2017) also provide certificates of robustness for neural networks against $\ell_\infty$ perturbations. That work uses SMT solvers, which are a tool from the formal verification literature. The SMT solver can answer the binary question *"Is there an adversarial example within distance $\epsilon$ of the input $x$?"*, and is correct whenever it terminates. The main drawback of SMT and similar formal verification methods is that they are slow—they have worst-case exponential-time scaling in the size of the network; moreover, to use them during training would require a separate search for each gradient step. Huang et al. (2017) use SMT solvers and are able to analyze state-of-the-art networks on MNIST, but they make various approximations such that their numbers are not true upper bounds.

Bastani et al. (2016) provide tractable certificates but require $\epsilon$ to be small enough to ensure that the entire $\ell_\infty$ ball around an input lies within the same linear region. For the networks and values of $\epsilon$ that we consider in our paper, we found that this condition did not hold. Recently, Hein & An-driushchenko (2017) proposed a bound for guaranteeing robustness to $\ell_p$-norm perturbations, based on the maximum $\frac{p}{p-1}$-norm of the gradient in the $\epsilon$-ball around the inputs. Hein & Andriushchenko (2017) show how to efficiently compute this bound for $p = 2$, as opposed to our work which focuses on $\ell_\infty$ and requires different techniques to achieve scalability.

Madry et al. (2017) perform adversarial training against PGD on the MNIST and CIFAR-10 datasets, obtaining networks that they suggest are "secure against first-order adversaries". However, this is based on an empirical observation that PGD is nearly-optimal among gradient-based attacks, and does not correspond to any formal robustness guarantee.

Finally, the notion of a *certificate* appears in the theory of convex optimization, but means something different in that context; specifically, it corresponds to a proof that a point is *near the optimum* of a convex function, whereas here our certificates provide upper bounds on *non-convex* functions. Additionally, while robust optimization (Bertsimas et al., 2011) provides a tool for optimizing objectives with robustness constraints, applying it directly would involve the same intractable optimization for $A_{\text{opt}}$ that we deal with here.

**Other approaches to verification.** While they have not been explored in the context of neural networks, there are approaches in the control theory literature for verifying robustness of dynamical systems, based on *Lyapunov functions* (Lyapunov, 1892; 1992). We can think of the activations in a neural network as the evolution of a time-varying dynamical system, and attempt to prove stability around a *trajectory* of this system (Tedrake et al., 2010; Tobenkin et al., 2011). Such methods typically use sum-of-squares verification (Papachristodoulou & Prajna, 2002; 2005; Parrilo, 2003) and are restricted to relatively low-dimensional dynamical systems, but could plausibly scale to

larger settings. Another approach is to construct families of networks that are provably robust a priori, which would remove the need to verify robustness of the learned model; to our knowledge this has not been done for any expressive model families.

**Adversarial examples and secure ML.** There has been a great deal of recent work on the security of ML systems; we provide only a sampling here, and refer the reader to Barreno et al. (2010), Biggio et al. (2014a), Papernot et al. (2016b), and Gardiner & Nagaraja (2016) for some recent surveys.

Adversarial examples for neural networks were first discovered by Szegedy et al. (2014), and since then a number of attacks and defenses have been proposed. We have already discussed gradient-based methods as well as defenses based on adversarial training. There are also other attacks based on, e.g., saliency maps (Papernot et al., 2016a), KL divergence (Miyato et al., 2015), and elastic net optimization (Chen et al., 2017); many of these attacks are collated in the `cleverhans` repository (Goodfellow et al., 2016). For defense, rather than making networks robust to adversaries, some work has focused on simply *detecting* adversarial examples. However, Carlini & Wagner (2017a) recently showed that essentially all known detection methods can be subverted by strong attacks.

As explained in Barreno et al. (2010), there are a number of different attack models beyond the test-time attacks considered here, based on different attacker goals and capabilities. For instance, one can consider *data poisoning* attacks, where an attacker modifies the training set in an effort to affect test-time performance. Newsome et al. (2006), Laskov & Šrndič (2014), and Biggio et al. (2014b) have demonstrated poisoning attacks against real-world systems.

**Other types of certificates.** Certificates of performance for machine learning systems are desirable in a number of settings. This includes verifying safety properties of air traffic control systems (Katz et al., 2017a;b) and self-driving cars (O'Kelly et al., 2016; 2017), as well as security applications such as robustness to training time attacks (Steinhardt et al., 2017). More broadly, certificates of performance are likely necessary for deploying machine learning systems in critical infrastructure such as internet packet routing (Winstein & Balakrishnan, 2013; Sivaraman et al., 2014). In robotics, certificates of stability are routinely used both for safety verification (Lygeros et al., 1999; Mitchell et al., 2005) and controller synthesis (Başar & Bernhard, 2008; Tedrake et al., 2010).

In traditional verification work, Rice's theorem (Rice, 1953) is a strong impossibility result essentially stating that most properties of most programs are undecidable. Similarly, we should expect that verifying robustness for arbitrary neural networks is hard. However, the results in this work suggest that it is possible to *learn* neural networks that are amenable to verification, in the same way that it is possible to write programs that can be formally verified. Optimistically, given expressive enough certification methods and model families, as well as strong enough specifications of robustness, one could even hope to train vector representations of natural images with strong robustness properties, thus finally closing the chapter on adversarial vulnerabilities in the visual domain.

**Reproducibility.** All code, data and experiments for this paper are available on the Codalab platform at https://worksheets.codalab.org/worksheets/0xa21e794020bb474d8804ec7bc0543f52/.

**Acknowledgements.** This work was partially supported by a Future of Life Institute Research Award and Open Philanthropy Project Award. JS was supported by a Fannie & John Hertz Foundation Fellowship and an NSF Graduate Research Fellowship. We are also grateful to Guy Katz, Zico Kolter and Eric Wong for providing relevant experimental results for comparison, as well as to the anonymous reviewers for useful feedback and references.

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

## A    DUALITY

In this section we justify the duality relation (14). Recall that the primal program is

$$\text{maximize } \langle M, P \rangle \tag{19}$$
$$\text{subject to } P \succeq 0, \text{diag}(P) \leq 1.$$

Rather than taking the dual directly, we first add the redundant constraint $\text{tr}(P) \leq d + m + 1$ (it is redundant because the SDP is in $d + m + 1$ dimensions and $\text{diag}(P) \leq 1$). This yields

$$\text{maximize } \langle M, P \rangle \tag{20}$$
$$\text{subject to } P \succeq 0, \text{diag}(P) \leq 1, \text{tr}(P) \leq d + m + 1.$$

We now form the Lagrangian for the constraints $\text{diag}(P) \leq 1$, leaving the other two constraints as-is. This yields the equivalent optimization problem

$$\text{maximize } \min_{c \geq 0} \langle M, P \rangle + c^\top (\mathbf{1} - \text{diag}(P)) \tag{21}$$
$$\text{subject to } P \succeq 0, \text{tr}(P) \leq d + m + 1.$$

Now, we apply minimax duality to swap the order of min and max; the value of (21) is thus equal to

$$\text{minimize } \max_{\substack{P \succeq 0, \\ \text{tr}(P) \leq d+m+1}} \langle M, P \rangle + c^\top (\mathbf{1} - \text{diag}(P)) \tag{22}$$
$$\text{subject to } c \geq 0.$$

The inner maximum can be simplified as

$$\mathbf{1}^\top c + (d+m+1) \cdot \left( \max_{P \succeq 0, \mathrm{tr}(P) \leq 1} \langle M - \mathrm{diag}(c), P \rangle \right) = \mathbf{1}^\top c + (d+m+1) \lambda_{\max}^+(M - \mathrm{diag}(c)). \quad (23)$$

Therefore, (22) simplifies to

$$\text{minimize } \mathbf{1}^\top c + (d+m+1) \lambda_{\max}^+(M - \mathrm{diag}(c)) \quad (24)$$
$$\text{subject to } c \geq 0.$$

This is almost the form given in (14), except that $c$ is constrained to be non-negative and we have $\mathbf{1}^\top c$ instead of $\mathbf{1}^\top \max(c, 0)$. However, note that for the $\lambda_{\max}^+$ term, it is always better for $c$ to be larger; therefore, replacing $c$ with $\max(c, 0)$ means that the optimal value of $c$ will always be non-negative, thus allowing us to drop the $c \geq 0$ constraint and optimize $c$ in an unconstrained manner. This finally yields the claimed duality relation (14).

