# OpenReview forum: "Certified Defenses against Adversarial Examples "
_ICLR.cc/2018/Conference — Accept (Poster)_

### Official Review · AnonReviewer3 · 2017-11-25
**A path towards a principled approach to train a robust classifier**

**Rating:** 8
**Confidence:** 4

**Review:**

This paper develops a new differentiable upper bound on the performance of classifier when the adversarial input in l_infinity is assumed to be applied.
While the attack model is quite general, the current bound is only valid for linear and NN with one hidden layer model, so the result is quite restrictive.

However the new bound is an "upper" bound of the worst-case performance which is very different from the conventional sampling based "lower" bounds. Therefore minimizing this upper bound together with a classification loss makes perfect sense and provides a theoretically sound approach to train a robust classifier.
This paper provides a gradient of this new upper bound with respect to model parameters so we can apply the usual first order optimization scheme to this joint optimization (loss + upper bound).
In conclusion, I recommend this paper to be accepted, since it presents a new and feasible direction of a principled approach to train a robust classifier, and the paper is clearly written and easy to follow.

There are possible future directions to be developed.

1. Apply the sum-of-squares (SOS) method.
The paper's SDP relaxation is the straightforward relaxation of Quadratic Program (QP), and in terms of SOS relaxation hierarchy, it is the first hierarchy. One can increase the complexity going beyond the first hierarchy, and this should provides a computationally more challenging but tighter upper bound.
The paper already mentions about this direction and it would be interesting to see the experimental results.

2. Develop a similar relaxation for deep neural networks.
The author already mentioned that they are pursuing this direction. While developing the result to the general deep neural networks might be hard, residual networks maybe fine thanks to its structure.

---

> ### Author Response · Authors · 2017-12-14
> **Comments on future directions**
>
> Thanks for the comments and thoughtful suggestions. Adding to the recommendations about the future work:
>
> 1. Sum-of-squares (SOS) method -- It is indeed interesting to check whether the higher degree of SOS gives us sufficient tightness to significantly improve the results. An obvious bottleneck in trying this out is the expensive computation. Given that our objective is similar to MAXCUT, for which it is currently unknown whether higher degree SOS relaxations give better approximation ratios, it is apriori unclear how much we could gain.
>
> 2. Develop a similar relaxation for deeper networks -- We agree with the reviewer that this is an interesting direction to pursue. In fact, we have already begun implementing an algorithm that works for arbitrary depth networks. As described in the response to reviewer 1, the basic idea is that the adversarial loss for arbitrary ReLU networks can be written as a non-convex quadratic program, which can then be relaxed to an SDP and trained with similar ideas to the present paper. As the reviewer mentions, it’s possible that resnets have additional structure that can be exploited efficiently, but our current proposal handles resnets as well.

---

### Official Review · AnonReviewer2 · 2017-11-27
**Innovative rigorous defense against adversarial attacks on NNs**

**Rating:** 8
**Confidence:** 4

**Review:**

The authors propose a new defense against security attacks on neural networks. The attack model involves a standard l_inf norm constraint. Remarkably, the approach outputs a security certificate (security guarantee) on the algorithm, which makes it appealing for security use in practice. Furthermore, the authors include an approximation of the certificate into their objective function, thus training networks that are more robust against attacks. The approach is evaluated for several attacks on MNIST data.

First of all, the paper is very well written and structured. As standard in the security community, the attack model is precisely formalized (I find this missing in several other ML papers on the topic). The certificate is derived with rigorous and sound math. An innovative approximation based on insight into a relation to the MAXCUT algorithm is shown. An innovative training criterion based on that certificate is proposed. Both the performance of the new training objective and the tightness of the cerificate are analyzed empirically showing that good agreement with the theory and good results in terms of robustness against several attacks.

In summary, this is an innovative paper that treats the subject with rigorous mathematical formalism and is successful in the empirical evaluation. For me, it is a clear accept. The only drawback I see is the missing theoretical and empirical comparison to the recent NIPS 2017 paper by Hein et al.

---

> ### Author Response · Authors · 2017-12-14
> **Comparison with recent NIPS 2017 paper by Hein et al.**
>
> Thanks for your interest in our work and the pointer to the relevant recent work by Hein and Andriushschenko. We have fixed this omission and include a discussion of the paper in the newest uploaded version of our work. To summarize the comparison: Firstly, their work focuses on perturbations in l-2 norm while ours considers the l-infty norm. Hence, there is no direct way to compare the experimental results. Theoretically, the general bound proposed for any l-p norm perturbation is similar to what we have in our work. However, the main challenge is to efficiently evaluate this bound. Hein and Andriushschenko show how to do this for p=2. In our work, we consider the attack model where p = \infty. This makes a significant difference in the computations involved.

---

### Official Review · AnonReviewer1 · 2017-11-28
**The main idea of  using upper bound (as opposed to lower bound) is reasonable. However, I find there are some limitations/weakness of the proposed method:**

**Rating:** 5
**Confidence:** 3

**Review:**

This paper derived an upper bound on adversarial perturbation for neural networks with one hidden layer. The upper bound is derived via (1) theorem of middle value; (2) replace the middle value by the maximum (eq 4); (3) replace the maximum of the gradient value (locally) by the global maximal value (eq 5); (4) this leads to a non-convex quadratic program, and then the authors did a convex relaxation similar to maxcut to upper bound the function by a SDP, which then can be solved in polynomial time.

The main idea of  using upper bound (as opposed to lower bound) is reasonable. However, I find there are some limitations/weakness of the proposed method:
1. The method is likely not extendable to more complicated and more practical networks, beyond the ones discussed in the paper (ie with one hidden layer)
2. SDP while tractable, would still require very expensive computation to solve exactly.
3. The relaxation seems a bit loose - in particular, in above step 2 and 3, the authors replace the gradient value by a global upper bound on that, which to me seems can be pretty loose.

---

> ### Author Response · Authors · 2017-12-14
> **Addressing concerns raised**
>
> Thank you for the comments! From our understanding of your review, it seems that there are three concerns, which we highlight and address below.
>
> 1. “The method is likely not extendable to more complicated and more practical networks, beyond the ones discussed in the paper (ie with one hidden layer)”
>
> Our general approach for obtaining networks with certified robustness can in fact extend to deeper networks. We have already begun implementing an algorithm that works for arbitrary depth networks. The basic idea is that the adversarial loss for arbitrary ReLU networks can be written as a non-convex quadratic program, which can then be relaxed to an SDP and trained with similar ideas to the present paper. We would be happy to give details if it would be helpful.
>
>
> 2. “SDP while tractable, would still require very expensive computation to solve exactly.”
>
> We would like to stress that we do not need to solve the SDP exactly. As discussed in Section 4 of our paper, our network can be trained via gradient descent on the dual. Even with inexact minimization of these dual variables, we get valid certificates. We acknowledge that training our model is slower than training regular networks, but it is not nearly as bad as if one had to exactly solve the SDP.
>
> We also note that at test time, the trained dual variables directly provide a certificate (with no extra computation) and hence checking robustness at test time is no slower than generating predictions for a regular network.
>
>
> 3. "The relaxation seems a bit loose - in particular, in above step 2 and 3, the authors replace the gradient value by a global upper bound on that, which to me seems can be pretty loose."
>
> An important insight from our experiments is the following: while our SDP bound can be quite loose on arbitrary networks, optimizing against this SDP certificate leads to networks where this certificate is substantially tighter (as seen in Figure 3). Minimizing the SDP upper bound forces the optimizer to avoid where the bound is loose, as such points have higher objective values. Hence, the general looseness of the relaxation does not impede the utility of the relaxation as a way of obtaining provably robust networks.

---

### Public Comment · ~Seong_Joon_Oh1 · 2017-11-09
**Discussion of a NIPS'17 paper**

Thanks for the interesting paper!
I wanted to leave a pointer to another NIPS'17 accepted paper:

https://arxiv.org/abs/1705.08475
Formal Guarantees on the Robustness of a Classifier against Adversarial Manipulation
Matthias Hein, Maksym Andriushchenko

Like the submission, Hein et al. also derive Lipschitz bound for neural networks with 1 hidden layer (see sec2.3) which looks similar to eq4&5 in the submission (modulo Lp norm used etc.). Indeed, this submission goes one step further to derive more bounds, but it would still be nice to discuss the difference.

Also small note: the main paper is 10 pages. In my view many intermediate inequalities (or proofs) can be deferred to appendix -- to highlight the main argument.

---

> ### Author Response · Authors · 2017-11-12
> **Discussion of a NIPS'17 paper**
>
> Hi Seong Joon Oh,
>
> Thanks for the pointer to the work by Matthias Hein, Maksym Andriushchenko. We'll cite this work in the next version of our paper. They propose a general bound on perturbations in the p-norm, that are necessary to cause misclassification, but only show how to compute this bound (for two layer neural networks) when p = 2.
> In our work, we consider the attack model where p = \infty. This makes a significant difference in the computations involved, where spectral-type bounds hold for p = 2; but optimizing over the L_\infty ball is typically more complex; we show how to efficiently do this in our work.
>
> Another key point of difference in that Hein and Andruishchenko use a "proxy" (Equation 6) for the actual lower bound in the proposed training algorithm.
> In general, there is no discussion on how this proxy relates to the actual derived bound (Equation 5). In our work, we propose a training algorithm that efficiently maximizes the lower bound on perturbation proposed (or equivalently, in the language of our paper, minimizing an upper bound on the adversarial loss).

---

### Public Comment · (anonymous) · 2017-11-12
**Comments/Questions**

Great work!

Some questions:
1. I'm curious -- how does the training time compare to that of Madry et. al.? Also, how much longer does this take than just normal training -- given that you have to compute the maximum eigen vector at each update?

2. Also, thoughts on generalizing this to multi-layered networks?

3. The Madry et. al. paper seems to consider 10^5 adv. examples or so, for training and attack. Are 5 random restarts sufficient to arrive at strong conclusions?

Given that the number of linear regions around a single point is fairly large, looks like 5 would be small. But, since the comparisons are fair (5 restarts for each defense), this seems very promising.

Also, how does regularization affect the accuracy?

---

### Author Response · Authors · 2017-11-14
**Addressing comments/questions**

Thanks for the comments and questions.

1. When run on the CPU, our training takes about 1.5 times the training time of Madry et al.
Scipy's max eigen vector computation runs on the CPU by default. We estimate that using a GPU implementation of Lancoz should also result in our training taking about 1.5 times the time taken by Madry et al. on the GPU. In general, our training is much slower than normal training. However, it's possible to speed up things using simple tricks. For example, using warm starts for the max eigen vector computations, by initializing with the solution of the previous iteration.

2. For multi-layered networks, optimizing for the worst-case adversarial example subject to ReLU constraints can be written as a different Quadratic Program (where the ReLU constraints are encoded by quadratic constraints). This Quadratic Program can then be relaxed to a semidefinite program, like in our paper. We are currently exploring  this idea empirically.

3. The Madry et al. paper considers experiments with 10^5 restarts to exhaustively understand the optimization landscape. The actual attacks use between 1 and 20 restarts. [Table 1 on page 12]. On the networks that we considered in our  paper, we didn't find any decrease in accuracy on increasing the number of random restarts beyond 5.

A really small value of reg parameter results in a network that has high clean test accuracy but low robustness, and similarly a very large value led to a network that had really low clean accuracy. However, for intermediate values of regularization, we observed that the classification loss (multiclass hinge) and regularization loss balance each other such that the worst case adversarial accuracy @ \eps = 0.1 remains nearly the same.  [the adversarial accuracy depends on the ratio of the multi class hinge loss to regularization loss which remains constant when the reg parameter is on the order of 0.05; we report results for this value in the paper].

---

### Public Comment · (anonymous) · 2017-12-03
**I think there was previous work on certifiable regions around an input.**

I am trying to understand the relation of this paper and

Measuring neural net robustness with constraints
by Bastani et al. NIPS 2016

In that paper, as far as I understand, the authors do the following: They are given some input image x1 and some neural network ( multiple layers, only ReLU activations) and lets say the output is also of the form sgn( w^T x ) so a linear function of the previous layer.
The paper obtains a polytope around x1 that provably will produce the same output as x1. Further they require that all the relus keep the same sign (which is a limitation but it critically allows them to get linear inequalities in x).

So they can now solve an LP to find the l_inf best possible epsilon that guarantees the same output label around x1.
In that sense this is a certifiable region around a given input, for multiple layer neural nets.

How does the authors' work compare ?

---

> ### Author Response · Authors · 2017-12-14
> **Comparison to work by Bastani et al., NIPS 2016**
>
> Thank you for your interest in our work! A crucial point of difference between the NIPS 2016 paper by Bastani et al., and our work is the following:  Bastani et al. provide certificates only for values of \epsilon that are small enough to ensure that the entire L-infty ball lies within the same linear region, i.e the ReLUs do not switch signs across the L-infty ball. For the networks and values of \epsilon we consider in our work, we found that most of the ReLUs cross signs and the L-infty balls do *not* lie within the same linear region. In contrast, our work provides certificates for all values of \epsilon.
>
> Another difference is that our training procedure minimizes a true upper bound on the adversarial loss, which is not the case in Bastani et al.’s work (like most other prior work).
>
> We have updated the discussion of related work to include this paper.

---

### Decision · Program_Chairs · 2018-01-29
**ICLR 2018 Conference Acceptance Decision**

**Decision:**

Accept (Poster)

**Comment:**

The paper presents a differentiable upper bound on the performance of classifier on an adversarially perturbed example (with small perturbation in the L-infinity sense). The paper presents novel ideas, is well-written, and appears technically sound. It will likely be of interest to the ICLR community.

The only downside of the paper is its limited empirical evaluation: there is evidence suggesting that defenses against adversarial examples that work well on MNIST/CIFAR do not necessarily transfer well to much higher-dimensional datasets, for instance, ImageNet. The paper could, therefore, would benefit from empirical evaluations of the defenses on a dataset like ImageNet.

---

> ### Public Comment · (anonymous) · 2018-05-18
> **Impressed even if they can do this on CIFAR**
>
> I've been trying to code their approach up, and it's not been fun. It's very difficult to even scale it to a modest sized problem. I've tried setting up a one layer problem on CIFAR with two classes but the slackness from the SDP relaxation just makes it impossible to train.
>
> Further, the breakdown after the eps of 0.1 is very fast even for MNIST.  If the authors have any comments to how to scale this, or make the relaxations not get bad every for very small eps (say 0.3 on MNIST), it would be very nice.